# Conditional Negative Sampling for Contrastive Learning of Visual Representations

**Mike Wu**[1], **Milan Mosse**[1,3], **Chengxu Zhuang**[2], **Daniel Yamins**[1,2], **Noah Goodman**[1,2]
Department of Computer Science[1], Psychology[2], and Philosophy[3]
Stanford University
{wumike, chengxuz, mmosse19, yamins, ngoodman}@stanford.edu

## Abstract

Recent methods for learning unsupervised visual representations, dubbed contrastive learning, optimize the noise-contrastive estimation (NCE) bound on mutual information between two transformations of an image. NCE typically uses randomly sampled negative examples to normalize the objective, but this may often include many uninformative examples either because they are too easy or too hard to discriminate. Taking inspiration from metric learning, we show that choosing semi-hard negatives can yield stronger contrastive representations. To do this, we introduce a family of mutual information estimators that sample negatives conditionally – in a "ring" around each positive. We prove that these estimators remain lower-bounds of mutual information, with higher bias but lower variance than NCE. Experimentally, we find our approach, applied on top of existing models (IR, CMC, and MoCo) improves accuracy by 2-5% absolute points in each case, measured by linear evaluation on four standard image benchmarks. Moreover, we find continued benefits when transferring features to a variety of new image distributions from the Meta-Dataset collection and to a variety of downstream tasks such as object detection, instance segmentation, and key-point detection.

## 1 Introduction

Supervised learning has given rise to human-level performance in several visual tasks (Russakovsky et al., 2015; He et al., 2017), relying heavily on large image datasets paired with semantic annotations. These annotations vary in difficulty and cost, spanning from simple class labels to more granular descriptions like bounding boxes and key-points. As it is impractical to scale high quality annotations, this reliance on supervision poses a barrier to widespread adoption. While supervised pretraining is still the dominant approach in computer vision, recent studies using unsupervised "contrastive" objectives, have achieved remarkable results in the last two years, closing the gap to supervised baselines (Wu et al., 2018; Oord et al., 2018; Hjelm et al., 2018; Zhuang et al., 2019; Hénaff et al., 2019; Misra & Maaten, 2020; He et al., 2019; Chen et al., 2020a;b; Grill et al., 2020).

Many contrastive algorithms are estimators of mutual information (Oord et al., 2018; Hjelm et al., 2018; Bachman et al., 2019), capturing the intuition that a good low-dimensional "representation" is one that linearizes the useful information embedded within a high-dimensional data point. In vision, these estimators maximize the similarity of encodings for two augmentations of the same image. This is trivial (e.g. assign all image pairs maximum similarity) unless the similarity function is normalized. This is typically done by comparing an image to "negative examples", which a model must assign low similarity to. We hypothesize that how we choose these negatives greatly impacts the representation quality. With harder negatives, the encoder is encouraged to capture more granular information that may improve performance on downstream tasks. While research in contrastive learning has explored architectures, augmentations, and pretext tasks, there has been little attention given to the negative sampling procedure. Meanwhile, there is a rich body of work in deep metric learning showing semi-hard negative mining to improve the efficacy of triplet losses. Inspired by this, we hope to bring harder negative sampling to modern contrastive learning.

Naively choosing difficult negatives may yield an objective that no longer bounds mutual information, removing a theoretical connection that is core to contrastive learning and has been shown to

be important for downstream performance (Tian et al., 2020). In this paper, we present a new estimator of mutual information based on the popular noise-contrastive estimator (NCE) that supports sampling negatives from conditional distributions. We summarize our contributions below:

1. We prove our Conditional-NCE (CNCE) objective to lower bound mutual information. Further, we show that although CNCE is a looser bound than NCE, it has lower variance. This motivates its value for representation learning.

2. We use CNCE to generalize contrastive algorithms that utilize a memory structure like IR, CMC, and MoCo to sample semi-hard negatives in just a few lines of code and minimal compute overhead.

3. We find that the naive strategy of sampling hard negatives throughout training can be detrimental. We then show that slowly introducing harder negatives yields good performance.

4. On four image classification benchmarks, we find improvements of 2-5% absolute points. We also find consistent improvements (1) when transferring features to new image datasets and (2) in object detection, instance segmentation, and key-point detection.

## 2 BACKGROUND

We focus on exemplar-based contrastive objectives, where examples are compared to one another to learn a representation. Many of these objectives (Hjelm et al., 2018; Wu et al., 2018; Bachman et al., 2019; Tian et al., 2019; Chen et al., 2020a) are equivalent to NCE (Oord et al., 2018; Poole et al., 2019), a popular lower bound on the mutual information, denoted by $\mathcal{I}$, between two random variables. This connection is well-known and stated in several works (Chen et al., 2020a; Tschannen et al., 2019; Tian et al., 2020; Wu et al., 2020). To review, recall:

$$\mathcal{I}(X;Y) \geq \mathcal{I}_{\text{NCE}}(X;Y) = \mathbf{E}_{x_i,y_i \sim p(x,y)}\mathbf{E}_{y_{1:k} \sim p(y)}\left[\log \frac{e^{f_\theta(x_i,y_i)}}{\frac{1}{k+1}\sum_{j \in \{i,1:k\}} e^{f_\theta(x_i,y_j)}}\right] \quad (1)$$

where $x, y$ are realizations of two random variables, $X$ and $Y$, and $f_\theta : X \times Y \to \mathbf{R}$ is a similarity function. We call $y_{1:k} = \{y_1, \ldots y_k\}$ negative examples, being other realizations of $Y$.

Suppose the two random variables in Eq. 1 are both transformations of a common random variable $X$. Let $\mathcal{T}$ be a family of transformations where each member $t$ is a composition of cropping, color jittering, gaussian blurring, among others (Wu et al., 2018; Bachman et al., 2019; Chen et al., 2020a). We call a transformed input $t(x)$ a "view" of $x$. Let $p(t)$ denote a distribution over $\mathcal{T}$, a common choice being uniform. Next, introduce an encoder $g_\theta : X \to \mathbf{S}^{n-1}$ that maps an example to a $L_2$-normalized representation. Suppose we have a dataset $\mathcal{D} = \{x_i\}_{i=1}^n$ of $n$ values for $X$ sampled from a distribution $p(x)$. Then, the contrastive objective for the $i$-th example is:

$$\mathcal{L}(x_i) = \mathbf{E}_{t,t',t_{1:k} \sim p(t)}\mathbf{E}_{x_{1:k} \sim p(x)}\left[\log \frac{e^{g_\theta(t(x_i))^T g_\theta(t'(x_i))/\tau}}{\frac{1}{k+1}\sum_{j \in \{i,1:k\}} e^{g_\theta(t(x_i))^T g_\theta(t_j(x_j))/\tau}}\right] \quad (2)$$

where $\tau$ is a temperature. The equivalence of Eq. 2 to NCE is immediate given $f_\theta(x,y) = g_\theta(x)^T g_\theta(y)/\tau$. Maximizing Eq. 2 chooses an embedding that pulls two views of the same example together while pushing two views of distinct examples apart. A drawback to this framework is that the number of negatives $k$ must be large to faithfully approximate the true partition. In practice, $k$ is limited by memory. Recent innovations have focused on tackling this challenge:

**Instance Discrimination** (Wu et al., 2018), or IR, introduces a memory bank of $n$ entries to cache embeddings of each example throughout training. Since every epoch we observe each example once, the memory bank will save the embedding of the view of the $i$-th example observed last epoch in its $i$-th entry. Representations stored in the memory bank are removed from the automatic differentiation tape, but in return, we can choose a large $k$ by querying $M$. A follow up work, Contrastive Multiview Coding (Tian et al., 2019), or CMC, decomposes an image into two color modalities. Then, CMC sums two IR losses where the memory banks for each modality are swapped.

**Momentum Contrast** (He et al., 2019), or MoCo, observed that the representations stored in the memory bank grow stale, since possibly thousands of optimization steps pass before updating an entry again. So, MoCo makes two important changes. First, it replaces the memory bank with a

first-in first-out (FIFO) queue of size $k$. During each minibatch, representations are cached into the queue while the most stale ones are removed. Second, MoCo introduces a second (momentum) encoder $g'_{\theta'}$ as a copy of $g_\theta$. The primary encoder $g_\theta$ is used to embed one view of $x_i$ whereas the momentum encoder is used to embed the other. Again, gradients are not propagated to $g'_{\theta'}$.

In this work, we focus on contrastive algorithms that utilize a memory structure that we repurpose in Sec. 4 to efficiently sample hard negatives from. In Sec. 7, we briefly discuss generalizations to contrastive algorithms that do not use a memory structure.

## 3 CONDITIONAL NOISE CONTRASTIVE ESTIMATION

In NCE, the negative examples are sampled i.i.d. from the marginal distribution, $p(y)$. Indeed, the existing proof that NCE lower bounds mutual information (Poole et al., 2019) assumes this to be true. However, choosing negatives in this manner may not be the best choice for learning a good representation. For instance, prior work in metric learning has shown the effectiveness of semi-hard negative mining in optimizing triplet losses (Wu et al., 2017; Yuan et al., 2017; Schroff et al., 2015). We similarly wish to exploit choosing semi-hard negatives in NCE *conditional on the current example* but to do so in a manner that preserves the lower bound on mutual information.

In presenting the theory, we assume two random variables $X$ and $Y$, deriving a general bound; we will return to the contrastive learning setting in Sec. 4. To begin, in Eq. 1, suppose we sample negatives from a distribution $q(y|x)$ conditional on a value $x \sim p(x)$ rather than the marginal $p(y)$, which is independent of $X$. Ideally, we would like to freely choose $q(y|x)$ to be any distribution but not all choices preserve a bound on mutual information[1]. This does not, however, imply that we can only sample negatives from $p(y)$ (Poole et al., 2019; Oord et al., 2018). One of our contributions is to formally define a family of conditional distributions $\mathcal{Q}$ such that for any $q(y|x) \in \mathcal{Q}$, drawing negative examples from $q$ defines an estimator that lower bounds $\mathcal{I}(X;Y)$. We call this new bound the Conditional Noise Contrastive Estimator, or CNCE. We first prove CNCE to be a bound:

**Theorem 3.1.** *(The Conditional NCE bound) Define $d$-dimensional random variables $X$ and $Y$ by a joint distribution $p(x,y)$ and let $Y_1, ..., Y_k$ be i.i.d. copies of $Y$ with the marginal distribution $p(y)$. Fix any function $f : (X, Y) \to \mathbf{R}$, any realization $x$ of $X$, and let $c = \mathbf{E}_{y \sim p(y)}[e^{f(x,y)}]$, the expected exponentiated similarity. Pick a set $B \subset \mathbf{R}$ strictly lower-bounded by $c$. Assume the pulled back set $S_B = \{y | e^{f(x,y)} \in B\}$ has non-zero probability (i.e. $p(S_B) > 0$). For $A_1, \ldots, A_k$ in the Borel $\sigma$-algebra over $\mathbf{R}^d$, define $A = A_1 \times \ldots \times A_k$ and let*

$$q((Y_1, \ldots, Y_k) \in A | X = x) = \prod_{j=1}^{k} p(A_j | S_B).$$

*Let $\mathcal{I}_{CNCE}(X;Y) = \mathbf{E}_{x,y \sim p(x,y)} \mathbf{E}_{y_1,\ldots,y_k \sim q(y_1,\ldots,y_k|x)} \left[ \log \frac{e^{f(x,y)}}{\frac{1}{k} \sum_{j=1}^{k} e^{f(x,y_j)}} \right]$. Then $\mathcal{I}_{CNCE} \leq \mathcal{I}_{NCE}$.*

*Proof.* To show $\mathcal{I}_{\text{CNCE}} \leq \mathcal{I}_{\text{NCE}}$, we show $\mathbf{E}_p[\log \sum_{j=1}^{k} e^{f(x,y_j)}] < \mathbf{E}_q[\log \sum_{j=1}^{k} e^{f(x,y_j)}]$. To see this, apply Jensen's to the left-hand side of $\log \mathbf{E}_p[\sum_{j=1}^{k} e^{f(x,y_j)}] < \log \sum_{j=1}^{k} e^{f(x,y_j)}$, which holds if $y_j \in S_B$ for $j = 1, \ldots, k$, and then take the expectation $\mathbf{E}_q$ of both sides. The last inequality holds by monoticity of $\log$, linearity of expectation, and the fact that $\mathbf{E}_p[e^{f(x,y_j)}] \leq e^{f(x,y_j)}$. $\qquad \square$

**Theorem Intuition.** For intuition, although using arbitrary negative distributions in NCE does not bound mutual information, we have found a restricted class of distributions $\mathcal{Q}$ where every member $q(y|x)$ "subsets the support" of the distribution $p(y)$. That is, given some fixed value $x$, we have defined $q(y|x)$ to constrain the support of $p(y)$ to a set $S_B$ whose members are "close" to $x$ as measured by the similarity function $f$. For every element $y \in S_B$, the distribution $q(y|x)$ wants to assign to it the same probability as $p(y)$. However, as $q(y|x)$ is not defined outside of $S_B$, we must renormalize it to sum to one (hence $p(A_j|S_B) = \frac{p(A_j \cap S_B)}{p(S_B)}$). Intuitively, $q(y|x)$ cannot change $p(y)$ too much: it must redistribute mass proportionally. The primary distinction then, is the smaller

---

[1] We provide a counterexample in Sec. A.1.

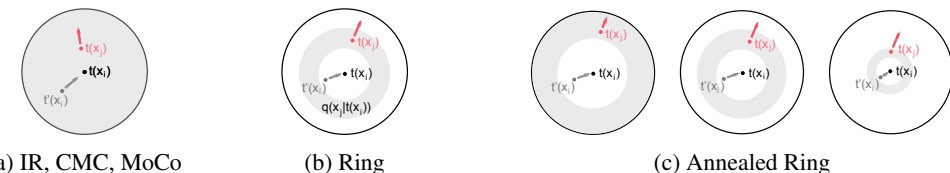

(a) IR, CMC, MoCo          (b) Ring          (c) Annealed Ring

Figure 1: Visual illustration of Ring Discrimination. Black: view of example $x_i$; gray: second view of $x_i$; red: negative samples; gray area: distribution $q(x|t(x_i))$. In subfigure (c), the negative samples are annealed to be closer to $t(x_i)$ through training. In other words, the support of $q$ shrinks.

support of $q(y|x)$, which forces samples from it to be harder for $f$ to distinguish from $x$. Thm. 3.1 shows that substituting $q(y|x)$ for $p(y)$ in NCE still bounds mutual information.

**Theorem Example 3.1.** We give a concrete example for the choice $B$ that will be used in Sec. 4. For any realization $x$, suppose we define two similarity thresholds $\omega_\ell, \omega_u \in \mathbf{R}$ where $c < \omega_\ell < \omega_u$. Then, choose $B = [w_\ell, w_u]$. In this case, the set $S_B$, which defines the support of the distribution $q(y|x)$, contains values of $y$ that are not "too-close" to $x$ but not "too-far". In contrastive learning, we might pick these similarity thresholds to vary the difficulty of negative samples.

Interestingly, Thm. 3.1 states that CNCE is looser than NCE, which raises the question: *when is a looser bound useful?* In reply, we show that while CNCE is a more biased estimator than NCE, in return it has lower variance. Intuitively, because $q(y|x)$ is the result of restricting $p(y)$ to a smaller support, samples from $q(y|x)$ have less opportunity to deviate, hence lower variance. Formally:

**Theorem 3.2.** *(Bias and Variance Tradeoff) Pick any $x, y \sim p(x, y)$. Fix the distribution $q(y_{1:k}|x)$ as stated in Theorem 3.1. Define a new random variable $Z(y_{1:k}) = \log\left(\frac{e^{f(x,y)}}{\frac{1}{k}\sum_{j=1}^{k} e^{f(x,y_j)}}\right)$ representing the normalized similarity. By Theorem 3.1, the expressions $\mathbf{E}_{p(y_{1:k})}[Z]$ and $\mathbf{E}_{q(y_{1:k}|x)}[Z]$ are estimators for $\mathcal{I}(X; Y)$. Suppose that the set $S_B$ is chosen to ensure $\text{Var}_{q(y_{1:k}|x)}[Z] \leq \text{Var}_{\tilde{q}(y_{1:k}|x)}[Z]$, where $\tilde{q}(A) = p(A|\text{ complement of } S_B)$. That is, we assume the variance of the normalized similarity when using $y_{1:k} \in S_B$ is smaller than when using $y_{1:k} \notin S_B$. Then $\text{Bias}_{p(y_{1:k})}(Z) \leq \text{Bias}_{q(y_{1:k}|x)}(Z)$ and $\text{Var}_{p(y_{1:k})}(Z) \geq \text{Var}_{q(y_{1:k}|x)}(Z)$.*

The proof can be found in Sec. A.2. Thm. 3.2 provides one answer to our question of looseness. In stochastic optimization, a lower variance objective may lead to better local optima. For representation learning, using CNCE to sample more difficult negatives may (1) encourage the representation to distinguish fine-grained features useful in transfer tasks, and (2) provide less noisy gradients.

## 4 RING DISCRIMINATION

We have shown CNCE to be a new bound on the mutual information that uses hard negative samples. Now we wish to apply CNCE to contrastive learning where the two random variables are again transformations of a single variable $X$. In this setting, for a fixed $x_i \sim p(x)$, the CNCE distribution is written as $q(x|t(x_i))$ for some transform $t \in \mathcal{T}$. Samples from $x \sim q(x|t(x_i))$ will be such that the exponentiated distance, $\exp\{g_\theta(t(x_i))^T g_\theta(t'(x))\}$, is at least a minimum value $c$. As in Example 3.1, we will choose $B = [\omega_\ell, \omega_u]$, a closed interval in $\mathbf{R}$ defined by two thresholds.

**Picking thresholds.** We pick the thresholds conditioned on the $i$-th example in the dataset, hence each example has a different set $B$. We first describe how to pick the upper threshold $\omega_u$. Given the $i$-example $x_i$, we pick a number $u \in [0, 100]$ representing an upper "percentile". We consider each example $x$ in the dataset to be in the support $S_B$ if and only if the (exponentiated) distance between the embedding of $x_i$ and $x$, or $\exp\{g_\theta(t(x_i))^T g_\theta(t'(x))\}$, is below the $u$-th percentile for all $x \in \mathcal{D}$. Call this maximum distance $\omega_u$. In other words, we construct $q(x|t(x_i))$ such that we ignore examples from the dataset whose embedding dot producted with the embedding of $x_i$ is above $\omega_u$. (Note that $u = 100$ recovers NCE.) For a small enough choice of $u$, the upper similarity threshold $\omega_u$ will be greater than $c$ (defined in Thm. 3.1 as the expected distance with respect to $p(x)$), and the samples from $q(x|t(x_i))$ will be harder negatives to discriminate from $x_i$.

In picking the *lower* threshold $\omega_\ell$, one could choose it to be 0, so $B = [0, \omega_u)$. However, picking the closest examples to $t(x_i)$ as its negative examples may be inappropriate, as these examples might be better suited as positive views rather than negatives (Zhuang et al., 2019; Xie et al., 2020). As an extreme case, if the same image is included in the dataset twice, we would not like to select it as a negative example for itself. Furthermore, choosing negatives "too close" to the current instance may result in representations that pick up on fine-grain details only, ignoring larger semantic concepts. This suggests removing examples from $q(x|t(x_i))$ we consider "too close" to $x_i$. To do this, we pick a lower percentile $0 \le \ell < u$. For each example $x \in \mathcal{D}$, we say it is in $S_B$ if $\exp\{g_\theta(t(x_i))^T g_\theta(t'(x))\}$ is below $\omega_u$ *and* also if it is above the $\ell$-th percentile of all distances with respect to $\mathcal{D}$. Call this minimum distance $\omega_\ell$. Fig. 2 visualizes this whole procedure.

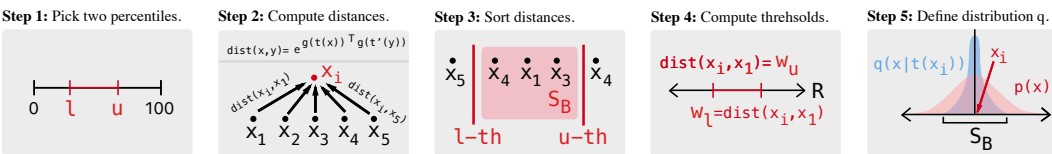

Figure 2: Defining the CNCE distribution $q(x|t(x_i))$. By choosing a lower and upper percentile $\ell$ and $u$, we implicitly define similarity thresholds $\omega_\ell$ and $\omega_u$ to construct a support of valid negative examples, $S_B$, which in turn, defines the distribution $q(x|t(x_i))$.

**Algorithm 1:** MoCoRing

```
# g_q, g_k: encoder networks
# m: momentum; t: temperature
# u: ring upper percentile
# l: ring lower percentile
tx1=aug(x)  # random augmentation
tx2=aug(x)
emb1=norm(g_q(tx1))
emb2=norm(g_k(tx2)).detach()
dps=sum(tx1*tx2)/t  # dot product
# sort from closest to farthest neg
all_dps=sort(emb1@queue.T/t)
# find indices of thresholds
ix_l=l*len(queue)
ix_u=u*len(queue)
ring_dps=all_dps[:,ix_l:ix_u]
# nonparametric softmax
loss=-dps+logsumexp(ring_dps)
loss.backward()
step(g_q.params)
# moco updates
g_k.params = m*g_k.params+\
             (1-m)*g_q.params
enqueue(queue,emb2); dequeue(queue)
# threshold updates
anneal(w_l); anneal(w_u)
```

**Ring Discrimination.** Having defined $\omega_\ell$ and $\omega_u$, we have a practical method of choosing $B$, and thus $S_B$ to define $q(x|t(x_i))$ for $i$-th example. Intuitively, we construct a conditional distribution for negative examples that are (1) not too easy since their representations are fairly similar to that of $x_i$ and (2) not too hard since we remove the "closest" instances to $x_i$ from $S_B$. We call this algorithm *Ring Discrimination*, or Ring, inspired by the shape of negative set (see Fig. 1).

Ring can be easily added to popular contrastive algorithms. For IR and CMC, this amounts to simply sampling entries in the memory bank that fall within the $\ell$-th to $u$-th percentile of all distances to the current example view (in representation space). Similarly, for MoCo, we sample from a subset of the queue (chosen to be in the $\ell$-th to $u$-th percentile), preserving the FIFO ordering. In our experiments, we refer to these as IRing, CM-CRing, MoCoRing, respectively. Alg. 1 shows PyTorch-like pseudocode for MoCoRing. One of the strengths of this approach is the simplicity: the algorithm requires only a few lines of code on top of existing implementations.

**Annealing Policy.** Naively using hard negatives can collapse to a poor representation, especially if we choose the upper threshold, $\omega_u$, to be very small early in training. At the start of training, the encoder $g_\theta$ is randomly initialized and cannot guarantee that elements in the $\ell$-th to $u$-th percentile are properly calibrated: if the representations are near random, choosing negatives that are close in embedding distance may detrimentally exclude those examples that are "actually" close. This could lock in poor local minima. To avoid this, we propose to use an annealing policy that reduces $\omega_u$ (and thus the size of the support $S_B$) throughout training. Early in training we choose $\omega_u$ to be large. Over many epochs, we slowly decrease $\omega_u$ closer to $\omega_l$, thereby selecting more difficult negatives. We explored several annealing policies and found a linear schedule to be well-performing and simple (see Sec. G). In our experiments, annealing is shown to be crucial: being too aggressive with negatives early in training produced representations that performed poorly on downstream tasks.

## 5 EXPERIMENTS

We explore our method applied to IR, CMC, and MoCo in four commonly used visual datasets. As in prior work (Wu et al., 2018; Zhuang et al., 2019; He et al., 2019; Misra & Maaten, 2020; Hénaff

et al., 2019; Kolesnikov et al., 2019; Donahue & Simonyan, 2019; Bachman et al., 2019; Tian et al., 2019; Chen et al., 2020a), we evaluate each method by linear classification on frozen embeddings. That is, we optimize a contrastive objective on a pretraining dataset to learn a representation; then, using a transfer dataset, we fit logistic regression on representations only. A better representation would contain more "object-centric" information, thereby achieving a higher classification score.

**Training Details.** We pick the upper percentile $u = 10$ and the lower percentile $\ell = 1$ although we anneal $u$ starting from 100. We resize input images to be 256 by 256 pixels, and normalize them using dataset mean and standard deviation. The temperature $\tau$ is set to 0.07. We use a composition of a 224 by 224-pixel random crop, random color jittering, random horizontal flip, and random grayscale conversion as our augmentation family $\mathcal{T}$. We use a ResNet-18 encoder with a output dimension of 128. For CMC, we use two ResNet-18 encoders, doubling the number of parameters. For linear classification, we treat the pre-pool output (size $512 \times 7 \times 7$) after the last convolutional layer as the input to the logistic regression. Note that this setup is equivalent to using a linear projection head (Chen et al., 2020a;b). In pretraining, we use SGD with learning rate 0.03, momentum 0.9 and weight decay 1e-4 for 300 epochs and batch size 256 (128 for CMC). We drop the learning rate twice by a factor of 10 on epochs 200 and 250. In transfer, we use SGD with learning rate 0.01, momentum 0.9, and no weight decay for 100 epochs without dropping learning rate. These hyperparameters were taken from Wu et al. (2018) and used in all of Table 1 for a consistent comparison. We found normalizing hyperparameters to be important for a fair comparison as many competing algorithms use different hyperparameters. For a state-of-the-art comparison, see Table 5.

| Model | Linear Evaluation | Model | Linear Evaluation | Model | Linear Evaluation | Model | Linear Evaluation |
|---|---|---|---|---|---|---|---|
| IR | 81.2 | IR | 60.4 | IR | 61.4 | IR | 43.2 |
| IRing | 83.9 (+2.7) | IRing | **62.3** (+1.9) | IRing | 64.3 (+2.9) | IRing | 48.4 (+5.2) |
| CMC* | 85.6 | CMC* | 56.0 | CMC* | 63.8 | CMC* | 48.2 |
| CMCRing* | **87.6** (+2.0) | CMCRing* | 56.0 (+0.0) | CMCRing* | **66.4** (+2.6) | CMCRing* | **50.4** (+2.2) |
| MoCo | 83.1 | MoCo | 59.1 | MoCo | 63.8 | MoCo | 52.8 |
| MoCoRing | 86.1 (+3.0) | MoCoRing | 61.5 (+2.4) | MoCoRing | 65.2 (+1.4) | MoCoRing | 54.6 (+1.8) |
| LA | 83.9 | LA | 61.4 | LA | 63.0 | LA | 48.0 |
| (a) CIFAR10 | | (b) CIFAR100 | | (c) STL10 | | (d) ImageNet | |

Table 1: Comparison of contrastive algorithms on four image domains. Superscript (*) indicates models that use twice as many parameters as others e.g. CMC has "L" and "ab" encoders.

The results for CIFAR10, CIFAR100, STL10, and ImageNet are in Table 1. Overall, IR, CMC, and MoCo all benefit from using more difficult negatives as shown by 2-5% absolute points of improvement across the four datasets. While we find different contrastive objectives to perform best in each dataset, the improvements from Ring are consistent: the Ring variant outperforms the base for every model and every dataset. We also include as a baseline Local Aggregation, or LA (Zhuang et al., 2019), a popular contrastive algorithm (see Sec. H) that implicitly uses hard negatives without annealing. We find our methods to outperform LA by up to 4% absolute.

| Model | Linear Eval. |
|---|---|
| IR | 81.2 |
| IRing | 83.9 |
| IRing (No Anneal) | 81.4 |
| IRing ($\ell = 0$) | 82.1 |

(a) CIFAR10

| Model | Linear Eval. |
|---|---|
| IR | 43.2 |
| IRing | 48.4 |
| IRing (No Anneal) | 41.3 |
| IRing ($\ell = 0$) | 47.3 |

(b) ImageNet

Table 2: Lesioning the effects of annealing and choice of $\ell$.

**Ablations: Annealing and Upper Boundary.** Having found good performance with Ring Discrimination, we want to assess the importance of the individual components that comprise Ring. We focus on the annealing policy and the exclusion of very close negatives from $S_B$. Concretely, we measure the transfer accuracy of (1) IRing without annealing and (2) IRing with an lower percentile $\ell = 0$, thereby excluding no close negatives. That is, $S_B$ contains *all* examples in the dataset with representation similarity less than the $\omega_u$ (a "ball" instead of a "ring"). Table 2 compares these ablations to IR and full IRing on CIFAR10 and ImageNet classification transfer. We observe that both ablations result in worse transfer accuracy, with proper annealing being especially important to prevent convergence to bad minima. We also find even with $\ell = 0$, IRing outperforms IR, suggesting both removing negatives that are "too close" and "too far" contribute to the improved representation quality.

**Transferring Features.** Thus far we have only evaluated the learned representations on unseen examples from the training distribution. As the goal of unsu-

pervised learning is to capture *general* representations, we are also interested in their performance on new, unseen distributions. To gauge this, we use the same linear classification paradigm on a suite of image datasets from the "Meta Dataset" collection (Triantafillou et al., 2019) that have been used before in contrastive literature (Chen et al., 2020a). All representations were trained on CIFAR10. For each transfer dataset, we compute mean and variance from a training split to normalize input images, which we found important for generalization to new visual domains.

| Model | Aircraft | CUBirds | DTD | Fungi | MNIST | FashionMNIST | TrafficSign | VGGFlower | MSCOCO |
|---|---|---|---|---|---|---|---|---|---|
| IR | 40.9 | 17.9 | 39.2 | 2.7 | 96.9 | 91.7 | 97.1 | 68.1 | 52.4 |
| IRing | 40.6 (-0.3) | 17.9 (+0.0) | 39.5 (+0.3) | 3.4 (+0.7) | 97.8 (+0.9) | 91.6 (+0.1) | 98.8 (+1.7) | 68.5 (+0.4) | 52.5 (+0.1) |
| MoCo | 41.5 | 18.0 | 39.7 | 3.1 | 96.9 | 90.9 | 97.3 | 64.5 | 52.0 |
| MoCoRing | 41.6(+0.1) | 18.6 (+0.6) | 39.5 (-0.2) | 3.6 (+0.5) | 97.9 (+1.0) | 91.3 (+0.4) | 99.3 (+2.0) | 69.1 (+4.6) | 52.6 (+0.6) |
| CMC | 40.1 | 15.8 | 38.3 | 4.3 | 97.5 | 91.5 | 94.6 | 67.1 | 51.4 |
| CMCRing | 40.8 (+0.7) | 16.8 (+1.0) | 40.6 (+2.3) | 4.2 (-0.1) | 97.9 (+0.4) | 92.1 (+0.6) | 97.1 (+2.5) | 69.1 (+2.0) | 52.1 (+0.7) |
| LA | 41.3 | 17.8 | 39.0 | 2.3 | 97.2 | 92.3 | 98.2 | 66.9 | 52.3 |

Table 3: Transferring CIFAR10 embeddings to various image distributions.

We find in Table 3 that the Ring models are competitive with the non-Ring analogues, with increases in transfer accuracies of 0.5 to 2% absolute. Most notable are the TrafficSign and VGGFlower datasets in which Ring models surpass others by a larger margin. We also observe that IRing largely outperforms LA. This suggests the features learned with more difficult negatives are not only useful for the training distribution but may also be transferrable to many visual datasets.

**More Downstream Tasks.** Object classification is a popular transfer task, but we want our learned representations to capture holistic knowledge about the contents of an image. We must thus evaluate performance on transfer tasks such as detection and segmentation that require different kinds of visual information. We study four additional downstream tasks: object detection on COCO (Lin et al., 2014) and Pascal VOC'07 (Everingham et al., 2010), instance segmentation on COCO, and keypoint detection on COCO. In all cases, we employ embeddings trained on ImageNet with a ResNet-18 encoder. We base these experiments after those found in He et al. (2019) with the same hyperparameters. However, we use a smaller backbone (ResNet-18 versus ResNet-50) and we freeze its parameters instead of finetuning them. We adapt code from Detectron2 (Wu et al., 2019).

| | COCO: Object Detection | | | COCO: Inst. Segmentation | | | COCO: Keypoint Detection | | | VOC: Object Detection | | |
|---|---|---|---|---|---|---|---|---|---|---|---|---|
| Arch. | Mask R-CNN, $R_{18}$-FPN, 1x schedule | | | | | | R-CNN, $R_{18}$-FPN | | | Faster R-CNN, $R_{18}$-C4 | | |
| Model | $AP^{bb}$ | $AP^{bb}_{50}$ | $AP^{bb}_{75}$ | $AP^{mk}$ | $AP^{mk}_{50}$ | $AP^{mk}_{75}$ | $AP^{kp}$ | $AP^{kp}_{50}$ | $AP^{kp}_{75}$ | $AP^{bb}$ | $AP^{bb}_{50}$ | $AP^{bb}_{75}$ |
| IR | 8.6 | 19.0 | 6.6 | 8.5 | 17.4 | 7.4 | 34.6 | 63.0 | 32.9 | 5.5 | 14.5 | 3.3 |
| IRing | **10.9** | **22.9** | **8.7** | 11.0 | 20.9 | 9.6 | 37.2 | 66.1 | 35.7 | 7.6 | 20.3 | 4.4 |
| MoCo | 6.0 | 14.3 | 4.0 | 10.8 | 21.4 | 9.7 | 37.6 | 66.5 | 36.9 | 7.3 | 17.9 | 4.1 |
| MoCoRing | 9.4 | 20.3 | 7.6 | **12.0** | **22.9** | **10.8** | **38.7** | **67.7** | **37.9** | **8.0** | **22.1** | **4.8** |
| LA | 10.2 | 22.0 | 8.1 | 10.0 | 20.3 | 9.0 | 36.3 | 65.3 | 35.1 | 7.6 | 20.0 | 4.3 |

Table 4: Evaluation of ImageNet representations using four visual transfer tasks.

We find IRing outperforms IR by around 2.3 points in COCO object detection, 2.5 points in COCO Instance Segmentation, 2.6 points in COCO keypoint detection, and 2.1 points in VOC object detection. Similarly, MoCoRing finds consistent improvements of 1-3 points over MoCo on the four tasks. Future work can investigate orthogonal directions of using larger encoders (e.g. ResNet-50) and finetuning ResNet parameters for these individual tasks.

## 6 RELATED WORK

Several of the ideas in Ring Discrimination relate to existing work. Below, we explore these connections, and at the same time, place our work in a fast-paced and growing field.

**Hard negative mining.** While it has not been deeply explored in modern contrastive learning, negative mining has a rich line of research in the metric learning community. Deep metric learning utilizes triplet objectives of the form $\mathcal{L}_{\text{triplet}} = d(g_\theta(x_i), g_\theta(x_+)) - d(g_\theta(x_i), g_\theta(x_-) + \alpha)$ where $d$ is a distance function (e.g. $L_2$ distance), $x_+$ and $x_-$ are a positive and negative example, respectively, relative to $x_i$, the current instance, and $\alpha \in \mathbf{R}^+$ is a margin. In this context, several approaches pick

semi-hard negatives: Schroff et al. (2015) treats the furthest (in $L_2$ distance) example in the same minibatch as $x_i$ as its negative, whereas Oh Song et al. (2016) weight each example in the minibatch by its distance to $g_\theta(x_i)$, thereby being a continuous version of Schroff et al. (2015). More sophisticated negative sampling strategies developed over time. In Wu et al. (2017), the authors pick negatives from a fixed normal distribution that is shown to approximate $L_2$ normalized embeddings in high dimensions. The authors show that weighting by this distribution samples more diverse negatives. Similarly, HDC (Yuan et al., 2017) simultaneously optimizes a triplet loss using many levels of "hardness" in negatives, again improving the diversity. Although triplet objectives paved the way for modern NCE-based objectives, the focus on negative mining has largely been overlooked. Ring Discrimination, being inspired by the deep metric learning literature, reminds that negative sampling is still an effective way of learning stronger representations in the new NCE framework. As such, an important contribution was to do so while retaining the theoretical properties of NCE, namely in relation to mutual information. This, to the best of our knowledge, is novel as negative mining in metric learning literature was not characterized in terms of information theory.

That being said, there are some cases of negative mining in contrastive literature. In CPC (Oord et al., 2018), the authors explore using negatives from the same speaker versus from mixed speakers in audio applications, the former of which can be interpreted as being more difficult. A recent paper, InterCLR (Xie et al., 2020), also finds that using "semi-hard negatives" is beneficial to contrastive learning whereas negatives that are too difficult or too easy produce worse representations. Where InterCLR uses a margin-based approach to sample negatives, we explore a wider family of negative distributions and show analysis that annealing offers a simple and easy solution to choosing between easy and hard negatives. Further, as InterCLR's negative sampling procedure is a special case of CNCE, we provide theory grounding these approaches in information theory. Finally, a separate line of work in contrastive learning explores using neighboring examples (in embedding space) as "positive" views of the instance (Zhuang et al., 2019; Xie et al., 2020; Asano et al., 2019; Caron et al., 2020; Li et al., 2020). That is, finding a set $\{x_j\}$ such that we consider $x_j = t(x_i)$ for the current instance $x_i$. While this does not deal with negatives explicitly, it shares similarities to our approach by employing other examples in the contrastive objective to learn better representations. In the Appendix, we discuss how one of these algorithms, LA (Zhuang et al., 2019), implicitly uses hard negatives and expand the Ring family with ideas inspired by it.

**Contrastive learning.** We focused primarily on comparing Ring Discrimination to three recent and highly performing contrastive algorithms, but the field contains much more. The basic idea of learning representations to be invariant under a family of transformations is an old one, having been explored with self-organizing maps (Becker & Hinton, 1992) and dimensionality reduction (Hadsell et al., 2006). Before IR, the idea of instance discrimination was studied (Dosovitskiy et al., 2014; Wang & Gupta, 2015) among many pretext objectives such as position prediction (Doersch et al., 2015), color prediction (Zhang et al., 2016), multi-task objectives (Doersch & Zisserman, 2017), rotation prediction (Gidaris et al., 2018; Chen et al., 2019), and many other "pretext" objectives (Pathak et al., 2017). As we have mentioned, one of the primary challenges to instance discrimination is making such a large softmax objective tractable. Moving from a parametric (Dosovitskiy et al., 2014) to a nonparametric softmax reduced issues with vanishing gradients, shifting the challenge to efficient negative sampling. The memory bank approach (Wu et al., 2018) is a simple and memory-efficient solution, quickly being adopted by the research community (Zhuang et al., 2019; Tian et al., 2019; He et al., 2019; Chen et al., 2020b; Misra & Maaten, 2020). With enough computational resources, it is now also possible to reuse examples in a large minibatch and negatives of one another (Ye et al., 2019; Ji et al., 2019; Chen et al., 2020a). In our work, we focus on hard negative mining in the context of a memory bank or queue due to its computational efficiency. However, the same principles should be applicable to batch-based methods (e.g. SimCLR): assuming a large enough batch size, for each example, we only use a subset of the minibatch as negatives as in Ring. Finally, more recent work (Grill et al., 2020) removes negatives altogether, which is speculated to implicitly use negative samples via batch normalization (Ioffe & Szegedy, 2015); we leave a more thorough understanding of negatives in this setting to future work.

## 7 DISCUSSION

**Computational cost of Ring.** To measure the cost of CNCE, we compare the cost an epoch of training MoCo/IR versus MoCoRing/IRing on four image datasets. Table 5a reports the average

| Model | CIFAR10 (sec.) | ImageNet (min.) | |
|---|---|---|---|
| IR | $136.0 \pm 4$ | $43.9 \pm 1$ | |
| IRing | $141.1 \pm 5$ (**1.1x**) | $51.0 \pm 1$ | (**1.2x**) |
| MoCo | $318.4 \pm 16$ | $61.1 \pm 1$ | |
| MoCoRing | $383.4 \pm 12$ (**1.2x**) | $64.9 \pm 1$ | (**1.1x**) |

(a) Average Epoch Cost

| Dataset | Arch. | MoCo-v2 | MoCoRing-v2 | |
|---|---|---|---|---|
| CIFAR10 | ResNet-18 | 90.1 | **91.9** | (+1.8) |
| CIFAR10 | ResNet-50 | 92.4 | **94.1** | (+1.6) |
| CIFAR100 | ResNet-18 | 65.1 | **67.3** | (+2.2) |
| STL10 | ResNet-18 | 74.8 | **76.7** | (+1.9) |

(b) Comparison with SOTA

| Transfer Task | MoCo | MoCoRing | |
|---|---|---|---|
| LibriSpeech Spk. ID (Panayotov et al., 2015) | 95.5 | **96.6** | (+1.1) |
| AudioMNIST (Becker et al., 2018) | 87.4 | **91.3** | (+3.9) |
| Google Commands (Warden, 2018) | 38.5 | **41.4** | (+2.9) |
| Fluent Actions (Lugosch et al., 2019) | 36.5 | **36.8** | (+0.3) |
| Fluent Objects (Lugosch et al., 2019) | 41.9 | **44.1** | (+2.2) |
| Fluent Locations (Lugosch et al., 2019) | 60.9 | **63.9** | (+3.0) |

(c) Speech Extension

| Dataset | SimCLR | SimCLRing | |
|---|---|---|---|
| CIFAR10 | 88.9 | **89.3** | (+0.4) |
| CIFAR100 | 63.5 | **64.1** | (+0.6) |
| STL10 | 71.2 | **72.1** | (+0.9) |

(d) SimCLRing Extension

Table 5: Generalizations of Ring to a new modality (a) and a batch-based algorithm (b).

cost over 200 epochs. We observe that Ring models cost no more than 1.5 times the cost of standard contrastive algorithms, amounting to a difference of 3 to 7 minutes in ImageNet and 10 to 60 seconds in three other datasets per epoch. In the context of deep learning, we do not find the cost increases to be substantial. In particular, since (1) the memory structure in IR and MoCo allow us to store and reuse embeddings and (2) gradients are not propagated through the memory structure, the additional compute of Ring amounts to one matrix multiplication, which is cheap on modern hardware. We used a single Titan X GPU with 8 CPU workers, and PyTorch Lightning (Falcon et al., 2019).

**Comparison with the state-of-the-art.** Unlike the experiments in Sec. 5, we now choose the optimal hyperparameters for MoCo-v2 (Chen et al., 2020b) separately for CIFAR10, CIFAR100, and STL10. Table 5b compares MoCo-v2 and its CNCE equivalent, MoCoRing-v2 using linear evaluation. We observe comparable improvements as found in Table 1 even with optimal hyperparameters. Notably, the gains generalize to ResNet-50 encoders. Refer to Sec. F for hyperparameter choices.

**Generalization to other modalities.** Thus far, we have focused on visual representation learning, although the same ideas apply to other domains. To exemplify the generality of CNCE, we apply MoCoRing to learning speech representations. Table 5c reports linear evaluation on six transfer datasets, ranging from predicting speaker identity to speech recognition to intent prediction. We find significant gains of 1 to 4 percent over 4 datasets and 6 transfer tasks with an average of 2.2 absolute percentage points. See Sec. E for experimental details.

**Batch-based negative sampling.** In Ring, we assumed to have a memory structure that stores embeddings, which led to an efficient procedure of mining semi-hard negatives. However, another flavor of contrastive algorithms removes the memory structure entirely, using the examples in the minibatch as negatives of one another. Here, we motivate a possible extension of Ring to SimCLR, and leave more careful study to future work. In SimCLR, we are given a minibatch $M$ of examples. To sample hard negatives, as before, pick $\ell$ and $u$ as lower and upper percentiles. For every example $x_i$ in the minibatch, only consider the subset of the minibatch $\{x : x \subseteq M, \exp\{g_\theta(t(x_i))^T g_\theta(t'(x))\}$ in the $\ell$-th and $u$-th percentiles in $M\}$ as negative examples for $x_i$. This can be efficiently implemented as a matrix operation using an element-wise mask. Thus, we ignore gradient signal for examples too far or too close to $x_i$ in representation. As before, we anneal $u$ from 100 to 10 and set $\ell = 1$. Table 5d report consistent but moderate gains over SimCLR, showing promise but room for improvement in future research.

# 8 CONCLUDING REMARKS

To conclude, we presented a family of mutual information estimators that approximate the partition function using samples from a class of conditional distributions. We proved several theoretical statements about this family, showing a bound on mutual information and a tradeoff between bias and variance. Then, we applied these estimators as objectives in contrastive representation learning. In doing so, we found that our representations outperform existing approaches consistently across a spectrum of contrastive objectives, data distributions, and transfer tasks. Overall, we hope our work to encourage more exploration of negative sampling in the recent growth of contrastive learning.

ACKNOWLEDGMENTS

This research was supported by the Office of Naval Research grant ONR MURI N00014-16-1-2007. MW is supported by the Stanford Interdisciplinary Graduate Fellowship as the Karr Family Fellow.

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
