# OpenReview forum: "Conditional Negative Sampling for Contrastive Learning of Visual Representations"
_ICLR.cc/2021/Conference — ICLR 2021 Poster_

### Official Review · AnonReviewer2 · 2020-10-28
**Reasonable direction, but needs more improvements**

**Rating:** 5
**Confidence:** 4

**Review:**

This paper adopts semi-hard negative mining, a sampling strategy widely used for metric learning, for contrastive self-supervised learning. Specifically, the paper chooses the negative samples in the range of $[w_l, w_u]$ percentiles (close, but not too close) in terms of the normalized feature distance. As the initial representation is not informative, the paper anneals down the percentile range. This sampling strategy improves the contrastive learning methods (IR, CMC, MoCO).

The paper has some good points:
- Applying semi-hard negative mining for contrastive learning is reasonable.
- Discussion on the property of the proposed estimator, CNCE.
- Empirically validate the proposed method improves the contrastive learning methods.

However, the paper needs more improvements for both method and presentation.


**Concerns in method**

A. Choice of the hyperparameters $[w_l,w_u]$.

Choosing "close, but not too close" samples is ambiguous and may depend on datasets, networks, and training methods. Is there some principle to choose hyperparameters? I checked both the main text and appendix but could not find how the paper selected the hyperparameters for experiments.

B. Cost of the negative mining

Searching negative samples for each update is quite expensive. How much the training time increased compared to the vanilla contrastive learning methods? Providing the training trend curve of the vanilla model and negative mining (using the clock time as an x-axis) would be insightful. It would also be great to discuss how to reduce the cost, e.g., use approximated nearest neighborhood search.

C. Negative mining for the *batch* setting?

For a single sample of $x_i$, it is easy to find the semi-hard negative samples. However, how to construct the batch $\{x_i\}$ such that each sample is effective negatives for the other samples? The batch should contain diverse samples; it would be interesting to consider the determinantal point process or submodular optimization formulation.


**Concerns in presentation**

There are lots of imprecise or undefined terms, unclear or unkind expressions, and typos. Here are some examples:
- Eq. (1) assumes to use $k$ negative samples ($i \notin \{1,...,k\}$ for a positive sample $i$), but Theorem 3.1 assumes to use $k-1$ negative samples
- The definition of the CNCE estimator comes after the property of it (Theorem 3.1)
- The definition of $S_B$ comes after the property of it. Also, it would be kinder to say "Assume $p(S_B) > 0$ for $S_B = \sim$" to "Assume that the set of random variables $S_B := \sim$ has a non-zero probability, i.e., $p(S_B) > 0$"
- "For Borel $A$" $\to$ "For a Borel set $A$"
- "Figure 1:" $\to$ "Figure 1: Visual illustration of ring discrimination"
- In Algorithm 1, do tx1 and tx2 receives the same input $x$?
- The evaluation metric "Transfer Acc." is not defined. Also, the term can be confused with "transferring features". Why not use the standard terminology "linear evaluation"?
- In Table 1, "three" image domains $\to$ "four" image domains


**Other comments**

Tons of similar techniques are concurrently proposed. It would be informative to discuss the relation with those works.
- Contrastive Learning with Hard Negative Samples
- Are all negatives created equal in contrastive instance discrimination?
- Self-supervised representation learning via adaptive hard-positive mining
- What Should Not Be Contrastive in Contrastive Learning
- Contrastive Learning with Stronger Augmentations

Is the sentence "A better representation would contain more "object-centric" information, thereby achieving a higher classification score." has some logical/empirical supports? Does "good" representation (in terms of downstream task performance) have some relation (in both directions) with the "object-centric" representation?

---

> ### Author Response · Authors · 2020-11-17
> **Rebuttal Response**
>
> We thank the reviewer for their thought-provoking questions.
>
> > how are the $w_\ell$ and $w_u$ chosen?
>
> First we pick numbers $0 \leq \ell < u \leq 100$ to represent percentiles. For the current example $x_i$, we compute embedding distances between all $x \in \mathcal{D}$and $x_i$ (via memory bank). We sort and exponentiate the distances, and choose $w_\ell$ and $w_u$ to be the distances at the $\ell$-th and $u$-th percentiles.  In experiments, we initially choose $u$ to be 100 and decrease it over training to 10 with a linear schedule. We fix $\ell = 1$.
>
> > What is the run-time cost of searching for negative samples?
>
> The cost of searching for negative samples is not substantial as a happy consequence of the reliance on memory banks: (1) there is no need to re-compute embeddings and (2) we do not backpropagate through the search. Therefore, the cost of searching is a single dot product between two matrices. In Section 7 of our revision, we added timing experiments over 4 datasets. We find on average a 1.2x increase, which we judge acceptable in the context of deep learning.
>
> > How do you handle the batch setting?
>
> Great question! We focused on IR and MoCo as sampling negatives in this setting is both simple and effective. For algorithms like SimCLR, where the negatives are the other elements in the minibatch, the problem is more nuanced.  We believe this is a substantial addition and not within the scope of this paper. Further, as MoCo and SimCLR are both state-of-the-art, we believe improving MoCo already pushes the boundaries.
>
> However, we are also interested and tested an idea:
>
> Our initial thoughts were also to encourage diversity in the batch. However, we realized it is less straightforward: recall that in NCE, the examples $x_i$ are picked i.i.d. from the marginal $p(x)$ (see the first expectation in Eq. 1). Using CNCE does not change this. So, if we construct the batch ourselves based on $q$, the examples $x_i$ are no longer distributed according to $p(x)$.
>
> We opt for a simpler approach: for each element in the minibatch, we only treat a subset of the minibatch as its negatives. We choose this subset with a lower and upper percentile based on embedding distance to the current element.  We added new experiments for SimCLR + Ring in Section 7. We find consistent but modest gains in 3 datasets, leaving plenty of room for future improvements. In our opinion, even these modest gains are non-trivial as one might have expected using more negatives to always be better.
>
> > In Alg. 1, do $t_1(x)$ and $t_2(x)$ receive the same input?
>
> Yes, MoCo compares embeddings of two transformations of the same image in the numerator.
>
> > A more thorough comparison with related work would be useful.
>
> We thank the reviewer for the pointer to concurrent work. These papers were new to us and we found them interesting to read. We discuss two of the papers we found most relevant to our work.
>
> Contrastive Learning with Hard Negative Samples (Robinson et. al.): this paper studies sampling hard negatives in contrastive learning as well. However, the negative distribution, called $q$ in both papers, is parameterized differently. Let $p$ be the marginal distribution. Both papers reweight the probability assigned by $p$ to every example: in theirs, this is done with an exponential term while in ours, this is done by restricting the support.  I believe there are strengths and weaknesses to both approaches. Their approach is more easily applied to SimCLR. But their approach still samples negatives from $p$ and upweights harder ones. If for example, the training dataset contains only a small set of hard negatives, choosing iid from $p$ may not find any, despite reweighting. In contrast, Ring subsets the possible samples to a set of harder negatives, so every negative must be at least a minimum level of “difficulty”. Finally, it is not clear if their approach preserves a lower bound on I(X;Y).
>
> Are all negatives created equal in contrastive instance discrimination? (Cai et. al.): This work does a thorough experimentation showing that the easiest negatives can be discarded and the hardest negatives are harmful. In our Appendix E (Figure 2), we also show that using hard negatives early in training hurts the representation. This observation has also been studied in deep metric learning: Wu. et. al. 2017 uses this to motivate the necessity of “semi”-hard negatives. Even Robinson et. al. discusses this in Section 6.1. The algorithm in Cai et. al. is quite similar to MoCoRing. That being said, the two papers show complementary findings: they focus on efficiency of using fewer negative samples whereas we show that you can also learn stronger representations by annealing. Further, we provide theory to ground this type of approach in mutual information.
>
> To briefly mention the other three papers, we found them less relevant as they focused on finding better augmentations, which is a complementary problem to better negatives.

---

> > ### Author Response · Authors · 2020-11-22
> > **Continued Response**
> >
> > > What is the relation between "object-centric" representations and downstream tasks?
> >
> > It is a good question because it is not always the case that a good contrastive representation captures object properties. In general, this relationship is set by the downstream tasks. In vision, the popular downstream tasks such as classification, detection, or segmentation are all properties of objects in the image. (For a more thorough discussion in the object detection setting, we refer to “What makes instance discrimination good for transfer learning” by Zhao et al.) As such, we would expect a representation that is highly performing to contain information about object features in the image. We will change the language in the text to be more precise that this is specific to the visual downstream tasks commonly studied in the field.

---

> > ### Comment · AnonReviewer2 · 2020-11-24
> > **Thanks to the Response**
> >
> > I sincerely read the rebuttal and other reviewers' reviews.
> >
> > **Method/Results**
> >
> > My concerns on run-time cost and batch extension are mostly addressed as the paper mainly considers the approaches using a memory bank (e.g., IR/MoCo). I also appreciate the authors' efforts to extend for SimCLR by subsampling negatives from the mini-batch, though the improvement seems to be less significant compared to IR/MoCO.
> >
> > Despite the partial extension for SimCLR, I believe the paper should more emphasize the primary focus is IR/MoCo, as the assumption of memory bank definitely restricts the applicability of the algorithm (though MoCo is also state-of-the-art as the rebuttal claimed).
> >
> > Furthermore, the performance reported in Table 6 seems too low. While the paper reports 90% on CIFAR-10 and 64% on CIFAR-100 for SimCLR, other implementations (including the original paper) report 93% for CIFAR-10 and 70% for CIFAR-100 (e.g., see https://github.com/HobbitLong/SupContrast).
> >
> > As Reviewer 3 also highlighted, the proposed technique should be validated upon the carefully tuned baseline (remark that many meta-analyses discovered that the baseline method outperforms the sophisticated methods, with careful optimization).
> >
> > **Presentation**
> >
> > I found that the revised draft significantly improved the presentation. However, I'm not sure this rushing for the submission and revision during the discussion period is a good practice for OpenReview.
> >
> > **Overall**
> >
> > My initial rating was on the unclear applicability of the method (e.g., for SimCLR), and the concern is addressed as the rebuttal clarified the primary focus of the paper is IR/MoCo.
> >
> > However, there are still some remaining concerns:
> > 1. Reported baseline results are too low (also highlighted by R3).
> > 2. The method should clarify its assumption and limitation.
> > 3. Significant revision during the discussion period.
> >
> > Thus, I currently recommend rating 5 (mostly due to concern 1).

---

> > > ### Author Response · Authors · 2020-11-25
> > > **Thanks for the feedback!**
> > >
> > > We are glad that we were able to address your initial concerns (regarding efficiency and generalization). To discuss the additional concerns about baselines, we provide some thoughts below:
> > >
> > > > Reported baseline results are too low.
> > >
> > > This was also raised by R3 and we fully agree that it is also meaningful to show improvements with Ring on the best performing baseline setups. We showed new results with MoCo-V2 in a new  general post above and hope the reviewer can refer to the updated numbers. We tried to pick the best documented setup for MoCo-v2 on these three datasets and improved the baseline numbers to match expected ones. On top of these, we show that MoCo-v2-Ring still has a 1.6-2.1% gain, which we find significant.
> > >
> > > Second, we want to clarify that the results in Table 6 are for SimCLR with Resnet-18 whereas the results in https://github.com/HobbitLong/SupContrast are for Resnet-50. If we refer to another repo (https://github.com/AidenDurrant/SimCLR-Pytorch) which shares results for Resnet-18 SimCLR on CIFAR10, we find 88% linear evaluation, which is what we have in Table 6. Since Table 6 is more motivation for future contributions rather than our core results, we think Resnet-18 results should suffice. However, if the reviewer would prefer to have Resnet-50, we can do so in the final revision.
> > >
> > > > The method should clarify its assumption and limitation.
> > >
> > > Yes! We will make this assumption clear. I updated a new revision with some edits: (1) in the contributions bullet points, I narrowed our claims to "contrastive algorithms that utilize a memory structure", and (2) in the end of the background (before Section 3), I state clearly that we restrict ourselves to contrastive algorithms with a memory structure. Finally, the discussion with batch-based negative sampling should hopefully make it more clear that extending to SimCLR and the like is future work.
> > >
> > > > Significant revision during the discussion period.
> > >
> > > We are very grateful for the opportunity to discuss and revise, making our contributions more precise and our experiments detailed. Although our revisions were significant, we have not deviated greatly from our initial claims and have only done additional work to clarify and experimentally strengthen them.

---

### Official Review · AnonReviewer4 · 2020-10-29
**1947 Initial review**

**Rating:** 5
**Confidence:** 4

**Review:**



Briefing:

This is an interesting paper that discusses the negative sample mining in visual representation learning. The authors discuss the theory and method to conditionally select the negative samples based on the dot product of representations in noise constructive estimation (NCE). Their theory shows that the NCE with negative examples sampling from a conditional distribution q is lower bounded with mutual information, and the object has higher bias and lower variance. The authors also provide the method to construct the conditional distribution by picking a ring surface where the dot product of representations is bounded within percentiles of data.

#######################################################

Pros:


The topic of this paper is popular and interesting. The negative sample mining strategy in unsupervised representation learning is well discussed and found effective in recent research.

The experiments show the effectiveness of their method compared with the original NCE methods.

#######################################################

Cons:


This paper's writing quality is limited, and this makes some points that are not easy to understand.

Why introduce many such details about IR and MoCo? It seems they are not heavily related to the contribution of this paper.

The paper limits the exploration of NCE to image representation learning with transform functions. This is not clear in the introduction section and may make readers confused about the contribution and purpose of the introduction. Can this theory and strategy be extended to other representation problems?

The claim of Borel sets A in theory 3.1 is not clear enough. How would you actually define the A2 to Ak? The proof of 3.1 is also unclear, and the notations here are in a mess.

The motivation to choose the way to construct q is still unclear. Why chose the dot product as the condition to build q? What's the connection between the "ring discrimination" and theory 3.1? It seems the sample selection strategy is not well supported by the theory the authors claim.

#######################################################

Please address and clarify the cons above in the rebuttal period.

---

> ### Author Response · Authors · 2020-11-17
> **Rebuttal Response**
>
> We thank the reviewer for their clarifying questions and comments.
>
> > The background is too long and perhaps, unrelated.
>
> We agree and have shortened the background significantly, especially the IR and MoCo explanations.
>
> > Is the approach general or specific to applications in vision?
>
> We rewrote the introduction to cut out details not relevant to our contributions. We also explicitly listed our contributions at the end of Section 1 in the revision.
>
> To summarize briefly, we present negative sampling specifically for contrastive learning of visual representations. We also focus on a class of algorithms that optimize NCE under lossy image transformations that are basis for recent breakthroughs (Wu et al., 2018; Oord et al., 2018; Hjelm et al., 2018; Zhuang et al., 2019; Henaff et al., 2019; Misra & Maaten, 2020; He et al., 2019; Chen et al., 2020a;b; Grill et al., 2020).
>
> That being said, we also believe Ring Discrimination should generalize to other modalities where contrastive algorithms have been successful e.g. speech (Oord et al., 2018; Al-Tahan & Mohsenzadeh, 2020; Kharitonov et. al. 2020). To showcase this generality, we added a new suite of experiments in Section 7 that compare speech representations learned by MoCo and MoCoRing on LibriSpeech using 6 transfer tasks. We find significant gains of up to 4% from using MoCoRing. We believe this is strong evidence for the generality of our algorithm.
>
> >  How would you actually define the $A_1$ to $A_k$ in Theorem 3.1?
>
> Good catch! We rewrote the text of Theorem 3.1 to be more clear. In particular, the sets $A_1$ to $A_k$ are each from the Borel $\sigma$-algebra over $\mathbf{R}^d$ (assuming $d$-dimensional random variables).
>
> > Why choose the dot product as the condition to build $q$?
>
> In Theorem 3.1, we construct $q$ according to an arbitrary similarity function $f: X \times Y \rightarrow \mathbf{R}$. So for the general case, we are not limited to a dot product.
>
> When we apply CNCE to contrastive learning (e.g. Ring Discrimination), we purposely parameterize $f(x,y) = g(t(x))^T g(t'(y))$, the dot product between embeddings of transforms of two images, $x$ and $y$. We do this to align to prior work: IR, CMC, LA, MoCo, SimCLR and many more use dot-product between embeddings as the distance function. This is likely due to wanting to learn encoders that linearize the useful information in a domain.
>
> > How does Ring Discrimination connect to Theorem 3.1?
>
> If possible, we encourage the reviewer to read our revised Section 3 and 4.
>
> To clarify Theorem 3.1, suppose we have two random variables $X$ and $Y$ and fix a value $x$. The theorem asks us to choose a set $B$ of real-valued numbers. Then, the set $S_B$ contains all values of $y$ such that $e^{f(x,y)}$ is in $B$. We should think of $S_B$ as the support of the distribution we will construct. The choice of $B$ lets us decide how close we want the negative examples to be to $x$. Then we define a distribution $q(y|x)$ as equal to $p(y)$ for all y in $S_B$ but renormalized to sum to 1. The utility of Theorem 3.1 is to show that replacing $p(y)$ with $q(y|x)$ in NCE builds a new estimator that still lower bounds I(X; Y). We call this estimator CNCE.
>
> Ring is a special case of CNCE for contrastive learning. First, the similarity function is parameterized as $f(x,y) = g(t(x))^T g(t’(y))$. Second, we set $B = [w_\ell, w_u]$, a closed interval defined by two real-valued thresholds $w_\ell$ and $w_u$.  Now, $S_B =${$y: e^{g(t(x))^T g(t'(y))} \in B$}, the set of all $y$ whose exponentiated distance to $x$ is in $B$. So, defining $q(y|x)$ on $S_B$ will guarantee that samples will not be "too far" from x and also not "too close". This is precisely Ring Discrimination. So, Theorem 3.1 covers this choice of B although it is more flexible (e.g. B need not be contiguous).
>
> In code, we pick two numbers $0\leq \ell < u \leq 100$ representing percentiles. We compute exponentiated distances of each example in the dataset with x. Sorting these, we can find the distance at the $\ell$-th and $u$-th percentile and set these to be $w_\ell$ and $w_u$.

---

### Official Review · AnonReviewer1 · 2020-11-02
**A good paper that tackles a very important problem in unsupervised learning**

**Rating:** 7
**Confidence:** 4

**Review:**

# Summary

Inspired by the effectiveness of hard negative mining in deep metric learning, this papers focuses on the problem of negative mining in unsupervised learning under the contrastive setting. One of the problems in this scenario is that naively selecting difficult negatives may yield an objective that no longer bounds mutual information, which is the basis for many contrastive objectives such as the Noise Contrastive Estimator. To address this problem, this paper formally defines a family of conditional distributions where negatives can be drawn from (negatives are chosen conditional on the current instance), while maintaining a lower bound on the NCE and on mutual information, resulting in a new estimator dubbed Conditional NCE. It also shows that, even though it’s a looser bound than NCE, it also has lower variance, which may lead to better local optima. Finally, within this family of conditional distributions, the paper proposes the Ring model, which takes inspiration from semi-hard negative mining approaches, and that can be applied to state-of-the-art contrastive algorithms in order to sample harder negatives, resulting in better representations.


# Pros

The paper tackles a typically disregarded problem in contrastive learning: hard negative mining. It shows the importance of selecting difficult negatives to obtain stronger unsupervised representations.

The method and contributions are very well motivated and introduced. The paper is also very well written, and includes a thorough review of the background needed in order to understand the proposed approach.

The strongest contribution of the paper is the definition of this family of conditional distributions, the CNCE, and the proof that it remains a lower bound on NCE, preserving the relation of contrastive learning to mutual information. Not less important is the proof that even though CNCE is a lower bound on NCE, it has however lower variance, leading to better local optima.

The Ring model results in a simple method that can be applied to any contrastive algorithm resulting in a better representation that outperforms existing approaches by a significant margin. This is shown in a thorough and extensive experimental analysis, which includes several benchmarks, transfer tasks, and ablation studies to validate the different components of the proposed approach.


# Cons

A minor comment is that the Ring model takes a lot of inspiration from other works in deep metric learning, which might limit its novelty. The concept of semi-hard negative mining is well-know in the field of deep metric learning and has been studied in numerous papers that have proposed more sophisticated approaches. I understand however that this wasn’t the main focus of the paper, being mainly paving the road to the exploration of negative sampling in contrastive learning.


# Recommendation

My overall recommendation is accept. The paper not only tackles a very important and typically disregarded problem in contrastive learning, but does it in a way that opens up the door to future research on this topic. It is also interesting to see hard negative mining from the mutual information point of view, making sure that by adding these conditional distributions the contrastive objective still remains a lower bound.

---

> ### Author Response · Authors · 2020-11-17
> **Rebuttal Response**
>
> We thank the reviewer for the positive and useful review!
>
> > There has been a rich history of research in semi-hard negative mining from deep metric learning that is worth mentioning.
>
> In the new revision, we now mention metric learning in the abstract and introduction to make obvious our inspiration from its literature. In Section 6, we reference several works from deep metric learning that we found to be similar to our approach, e.g. Schroff et. al. (2015), Oh Song et. al. (2016), Wu et. al. (2017), Yuan  et. al. (2017), which describe semi-hard negative mining procedures. We draw inspirations from these approaches but find the application to modern contrastive learning to be novel and relevant for future work.

---

### Official Review · AnonReviewer3 · 2020-11-06
**Interesting direction, presentation and experiments need to be enhanced**

**Rating:** 6
**Confidence:** 4

**Review:**

This paper propose to sample effective hard negative samples in contrastive learning, conditioned on given anchor point. Authors proved that their new objective is a more biased lower bound than InfoNCE, but with less variance. Experiments on several datasets as well as transferring datasets show some promising results.

Pros:
- Conditional negative sampling in contrastive learning is currently less studied, and this paper starts on this direction.
- The relative improvement of the new objective function upon multiple popular contrastive learning methods have been verified.
- The strategy in final realization is easy to implemented (though this realization is not very closely coherent to their theoretical analysis). In the analysis, they specify expectation value c as bound, while in experiments, the tuning of w_l can be tricky and a bit hacky.
- The implementation can be simple, which I appreicaite.

Cons:
- The written is not very good. Specifically, I found the usage of p and q in section 3 is a bit hard to follow. Specifically, I did not follow the sentence, "The remaining probability mass assigned by p to elements outside S_B is renormalized to sum to one...", do you mean conditional p or marginal p, or else? Overall, the section 3 is not well clarified for me
- While the experimental explicitly remove "two closed" samples, this part is completely missing in the theoretical analysis. E.g., the authors did not discuss the cases that the c threshold in Theorem 3.1 is too high such that set B only contains positive samples.
- The authors do not prove, in experience, that whether such improvement still holds on more solid baselines with larger networks and recent contrastive learning tricks. E.g., how would MoCoRing works with MoCo v2 baseline with 800 epochs training of R-50 model on ImageNet?
- Specifically for MoCo which maintains a FIFO queue, I am not convinced that the improvement mainly comes from hard negatives. It might because it mainly comes from removing "too close" negatives which are false negatives. I am imagine the case that, if we have labels to remove all false negatives in MoCo for each anchor, then all remaining negatives are true negatives. Then, does it still help to remove easy negatives to certain extent, or it's better to keep all true negatives you have in the queue? This would be some interesting experiments to see.
- The results on MS COCO is way off. I do not believe that switching from R-50 to R-18 will lead such a significant drop in AP, e.g. AP of bounding box drops from ~38 to ~10.

Overall, I have a mixed feeling over this paper, and would appreciate authors response.

======= updated ======
The authors response partially addressed my concerns and I would raise the rating to 6.

---

> ### Author Response · Authors · 2020-11-17
> **Rebuttal Response**
>
> We thank the reviewer for their thoughtful comments.
>
> > The sentence "The remaining probability mass assigned by p to elements outside $S_B$ is renormalized to sum to one..." is unclear.
>
> Say we construct an initial distribution $q^*(y|x)$ to be equal to the marginal distribution, $p(y)$, for all $y \in S_B$ (we should think of $S_B$ as the support we want our constructed distribution to have). But $S_B $ is a subset of the support of $p(y)$ so $q^*(y|x)$ does not sum to 1. To make it a valid probability distribution, we need to normalize $q^*(y|x)$. Let $q(y|x) = q^*(y|x) / \sum_{y \in S_B} q^*(y|x)$. But this is the same as $q(y|x) = q^*(y|x) / p(S_B)$.
>
> > Removing negative examples that are “too close” to the current example is not covered by Theorem 3.1.
>
> This is a good question and a good opportunity for us to clarify Theorem 3.1 and the relationship of CNCE to Ring.
>
> To clarify Theorem 3.1, suppose we have two random variables $X$ and $Y$ and fix a value $x$. The theorem asks us to choose a set $B$ of real-valued numbers. Then, the set $S_B$ contains all values of $y$ such that $e^{f(x,y)}$ is in $B$. We should think of $S_B$ as the support of the distribution we will construct. The choice of $B$ lets us decide how close we want the negative examples to be to $x$. Then we define a distribution $q(y|x)$ as equal to $p(y)$ for all y in $S_B$ but renormalized to sum to 1. The utility of Theorem 3.1 is to show that replacing $p(y)$ with $q(y|x)$ in NCE builds a new estimator that still lower bounds I(X; Y). We call this estimator CNCE.
>
> Ring is a special case of CNCE for contrastive learning. First, the similarity function is parameterized as $f(x,y) = g(t(x))^T g(t’(y))$. Second, we set $B = [w_\ell, w_u]$, a closed interval defined by two real-valued thresholds $w_\ell$ and $w_u$.  Now, $S_B =${$y: e^{g(t(x))^T g(t'(y))} \in B$}, the set of all $y$ whose exponentiated distance to $x$ is in $B$. So, defining $q(y|x)$ on $S_B$ will guarantee that samples will not be "too far" from x and also not "too close". This is precisely Ring Discrimination. So, Theorem 3.1 covers this choice of B although it is more flexible (e.g. B need not be contiguous).
>
> In code, we pick two numbers $0\leq \ell < u \leq 100$ representing percentiles. We compute exponentiated distances of each example in the dataset with x. Sorting these, we can find the distance at the $\ell$-th and $u$-th percentile and set these to be $w_\ell$ and $w_u$.
>
> > What if the threshold $c$ is so high that the set $S_B$ only contains positive samples?
>
> Great question! Pick a value $x$. Recall that $c$ is the average exponentiated distance between $x$ and $y$ over all $y \sim p(y)$. So unless the data distribution is such that most examples are positive samples of $x$, it should not be the case that $c$ is so high that whatever $S_B$ we choose contains only positive samples. For example, given an image $x$ from ImageNet. For $S_B$ to only contain positive samples would mean that the “average image” is as close to $x$ in representation as an image of the same class as $x$.
>
> > Do experiments generalize to larger architectures e.g. R-50?
>
> Ideally, we would be able to show experiments on a diverse range of architectures and hyperparameters. However, fitting 800 epochs of R-50 is very computationally intensive and difficult for us to do. Instead, we hoped to showcase consistent improvements on different objectives, transfer tasks, and image distributions (along with the new speech and SimCLR experiments: see Section 7). Together, we found this convincing that the method is generalizable.
>
> > How do you compare the effects of removing close negatives versus using hard negatives.
>
> This is a great question! We actually had experiments comparing the importance of removing "too hard" negatives vs "too easy" negatives in Table 2 on page 6. The rows for "IRing" represent sampling negatives whose embedding distance to the current example is between the $\ell$-th and $u$-th percentile, thereby removing those in the $[0, \ell)$ and $(u, 100]$ percentiles. The rows for “IRing ($\ell$ = 0)” therefore sample negatives in the $[0, u]$ percentiles, keeping the ones closest to the current example. Thus, these two models directly compare the effect of removing “too close” negatives. We see gains from only removing “too far” negatives (by 0.9 points in CIFAR10 and 4.1 points in ImageNet). We see additional gains from also removing the “too close” negatives (by 1.8 points in CIFAR10 and 1.1 points on ImageNet). We can reasonably conclude that both play an important role, motivating why in our main experiments, we remove both “too far” and “too close” negatives.
>
> > MS COCO results seem too low.
>
> Table 4 reported use of a “frozen” R-18 encoder. That is, we do not finetune the parameters. While this results in lower performance, we do believe it to be a fair test of the quality of the representation in capturing object properties.

---

> > ### Author Response · Authors · 2020-11-18
> > **Added MoCo V2 Experiments**
> >
> > > Do the results generalize to MoCo V2?
> >
> > This is a good question as we claim our methods to be general. To test this, we compare MoCo-v2 and MoCoRing-v2 on CIFAR10, CIFAR100, and STL10. The changes from MoCo to MoCo-v2 are (1) using different image augmentations (e.g. gaussian blur and different color jittering), and (2) using a MLP projection head. No changes are needed to the Ring formulation.
> >
> > STL10 | MoCo-v2  | 62.3
> >
> > STL10 | MoCoRing-v2 | 64.0 (+1.7)
> >
> > CIFAR10 | MoCo-v2 | 84.7
> >
> > CIFAR10 | MoCoRing-v2 | 88.0 (+3.3)
> >
> > CIFAR100 | MoCo-v2 | 62.4
> >
> > CIFAR100 | MoCoRing-v2 | 65.6 (+3.2)
> >
> > We find consistent and significant improvements over MoCo-v2 that parallel the improvements found on MoCo. We note that MoCo-v2 / MoCoRing-v2 perform worse than MoCo / MoCoRing on STL10; we suspect this to be due to the smaller dataset size. For the next revision, we will add MoCo-v2 results for ImageNet, meta-dataset, and the object detection / segmentation experiments.

---

### Author Response · Authors · 2020-11-17
**Updated submission with revisions.**

We thank the reviewers for their positive feedback and important suggestions. We rewrote substantially for more clarity and simpler language. Below, we provide a “diff” of the main changes to text in the new revision.

- Tightened the introduction to be more clear on scope. Replaced general statements with more focus on motivation of improving visual contrastive algorithms. We include a list of contributions at the end of Section 1.
- Shortened background to focus on more essential details.
- Rewrote Section 3 for clarity. We do not unnecessarily abbreviate terms and try to be clear about which distributions we are referencing. We simplified the notation as well.
- Added an example below Theorem 3.1 to give the reader context.
- Rewrote Section 4 to clarify connection to Section 3.
- Mentioned the inspiration for our method from metric learning earlier in paper.
- Added Figure 2 as visual support for Section 4 (Ring Discrimination).
- Fixed many typos and simplified language.
- Added Section 7 with three sets of additional experiments: (1) timing costs comparing IR to IRing and MoCo to MoCoRing (we find increases of only 1.2x per epoch); (2) application of MoCoRing to learning speech representations (we find significant gains of up to 4% absolute over 6 transfer tasks); (3) application of Ring to SimCLR (where we find modest but consistent improvements).

We aim to address individual concerns below. We hope reviewers can use the revised text for reference.

---

> ### Author Response · Authors · 2020-11-18
> **Linear Evaluation with MoCo-V2**
>
> We bring an additional result to the attention of the reviewers. We compare the performance of MoCo-v2 and MoCoRing-v2 using linear evaluation on CIFAR10, CIFAR100, and STL10. We found improvements of 3.3%, 3.2%, 1.7% absolute points, respectively by using Ring. This is comparable to the benefits of MoCoRing over MoCo. Please refer to the response to AnonReviewer3 for more details.

---

### Author Response · Authors · 2020-11-25
**Updated MoCo-v2 Results**

Multiple reviewers rightfully addressed concerns that our baseline numbers did not match prior work due to a difference in setup. In our paper, we tried to use a consistent setup that ended up sacrificing performance on baselines. We agree that it is also important to show that gains from using Ring persist on top on the best performing setups.

As such, we studied the hyperparameters used in the SimCLR repos suggested by Reviewers 2 and 3, as well as the Pytorch Lightning Bolts repo (https://github.com/PyTorchLightning/pytorch-lightning-bolts) which has a MoCo-V2 implementation to match our setup to those described in the papers as best as possible. After several attempts and changes, we arrived at the following (new) results:

CIFAR10 ResNet18 | MoCo-v2 90.09 | MoCoRing-v2 91.96 (+1.87)

STL10 ResNet18 | MoCo-v2 74.81 | MoCoRing-v2 76.72 (+1.91)

CIFAR100 ResNet18 | MoCo-v2 65.18 | MoCoRing-v2 67.29 (+2.11)

Our baselines now more closely match prior work (~90% on CIFAR10, ~75% on STL10 and a 3% incease in CIFAR100 from our last results). Additionally, we find that using Ring still increases the performance by 1.6-2.1%. For the final revision, we will include these in the main paper.

Additionally, we know that using larger architectures is also important. As such, we show Resnet-50 results for CIFAR10:

CIFAR10 ResNet50 | MoCo-v2 92.47 | MoCoRing-v2 94.10 (+1.63)

The gain from using Ring remains significant! (We also note that these two models were only trained for 200 epochs and may see further increases given the time to finish 800 epochs.). Overall, we believe these results, along with the ones in our paper which standardize the setups, to be good evidence that the Ring algorithm is improving the representations in a meaningful manner.

Finally, we would like to thank the reviewers for taking the time to respond and provide continued feedback. We feel that our paper has become more clear and stronger in the rebuttal process and credit the reviewers for this.

---

### Decision · Program_Chairs · 2021-01-07
**Final Decision**

**Decision:**

Accept (Poster)

**Comment:**

This paper addresses the problem of how best to sample hard negatives during contrastive learning, a topic of importance for the recently resurgent field of metric learning / contrastive loss-based unsupervised representation learning. Backed by theoretical results for a new low-variance version of the NCE, the paper proposes an easy-to-implement "Ring" method for selecting negatives that are at just the right level of difficulty, neither too hard nor too easy.

Happily, this is a paper that has improved significantly through the interactive peer review of a dedicated set of reviewers combined with prompt responses from the authors. Perhaps the result that tipped this paper over the line in my assessment: the new experimental results now show significant gains from applying the "Ring" approach for hard negative sampling to near-state-of-the-art implementations of the MoCo-v2 approach, which is among the leading unsupervised visual feature learning approaches.